# Overcoming label shift in target-aware federated learning

**Edvin Listo Zec**[*]
*Research Institutes of Sweden RISE*

**Adam Breitholtz**[*]
*Chalmers University of Technology and the University of Gothenburg*

**Fredrik D. Johansson**                                      *fredrik.johansson@chalmers.se*
*Chalmers University of Technology and the University of Gothenburg*

**Reviewed on OpenReview:** *https: // openreview. net/ forum? id= dQAsAmb1Xb*

## Abstract

Federated learning enables multiple actors to collaboratively train models without sharing private data. Existing algorithms are successful and well-justified in this task when the intended *target domain*, where the trained model will be used, shares data distribution with the aggregate of clients, but this is often violated in practice. A common reason is label shift—that the label distributions differ between clients and the target domain. We demonstrate empirically that this can significantly degrade performance. To address this problem, we propose FedPALS, a principled and practical model aggregation scheme that adapts to label shifts *to improve performance in the target domain* by leveraging knowledge of label distributions at the central server. Our approach ensures unbiased updates under federated stochastic gradient descent which yields robust generalization across clients with diverse, label-shifted data. Extensive experiments on image classification tasks demonstrate that FedPALS consistently outperforms baselines by aligning model aggregation with the target domain. Our findings reveal that conventional federated learning methods suffer severely in cases of extreme label sparsity on clients, highlighting the critical need for target-aware aggregation as offered by FedPALS.

## 1 Introduction

Federated learning (FL) has emerged as a powerful paradigm for training machine learning models collaboratively across multiple clients without sharing data (McMahan et al., 2017; Kairouz et al., 2021). This is attractive in problems where privacy is paramount, such as healthcare applications (Sheller et al., 2020), finance (Byrd & Polychroniadou, 2020), and the fine-tuning of foundation models (Hilmkil et al., 2021). A pervasive challenge is that the performance of federated learning systems is harmed by systematic differences in the clients' data distributions. An important example of this is label shift (Zhao et al., 2018; Woodworth et al., 2020), in which the relative prevalence of classes varies between clients.

Most federated learning research, including works that address distributional shifts(Zhao et al., 2018; Li et al., 2019; Ramezani-Kebrya et al., 2023), focuses on what we term *standard* federated learning, in which the test distribution matches the combined, average distribution of training clients. This is rather arbitrary; the aggregate depends directly on the number of samples each client has collected and which clients are included in the federation at the time. Real-world applications often require generalization to a distinct *target* domain, different from the client aggregate and any single client. For example, in a retail application, multiple stores (clients) collaboratively train a sales prediction model using their local purchase histories. The trained model may be deployed in an updated federation with added or removed stores, or in a single

---

[*]Equal contribution

*new* store with different customer preferences. As we will see, the performance depends greatly on the choice of target distribution.

The problem of generalizing under distributional shifts has been extensively studied in centralized settings, often under the umbrella of domain adaptation (Blanchard et al., 2011; Ganin et al., 2016; Breitholtz & Johansson, 2022). However, traditional domain adaptation techniques, such as sample re-weighting (Lipton et al., 2018) or domain-invariant representation learning (Arjovsky et al., 2020), require access to data from both source and target domains. This requirement is incompatible with the decentralized nature of federated learning, where neither the server nor the clients share data between them. While several techniques have been proposed to address client heterogeneity in standard FL, such as regularization (Li et al., 2020a; 2021), clustering (Ghosh et al., 2020; Vardhan et al., 2024), and meta-learning (Chen et al., 2018; Jiang et al., 2019), they do not address generalization to a known target domain different from the client aggregate.

**Contributions**   We study federated learning under label shift with a target domain that differs from both the client-average distribution and any single client. To understand the limitations of standard approaches in this setting, we investigate *target-aware* learning, where the central server *knows* the label distribution of both the target domain and all clients (see Section 2). For this setting, we propose a novel aggregation scheme called FedPALS that optimizes a convex combination of client models to ensure that the aggregated model is better suited for the label distribution of the target domain (Section 3). We prove that the resulting stochastic gradient update behaves, in expectation, as centralized learning in the target domain (Proposition 1), and examine its relation to standard federated averaging (Proposition 3.2). We demonstrate the limitations of standard methods in the targeted setting and the effectiveness of FedPALS through an extensive empirical evaluation (Section 5), showing that it outperforms traditional approaches in scenarios where distributional shifts pose significant challenges, at the small cost of sharing client label marginals with the central server. Moreover, we observe that traditional methods struggle particularly in scenarios where training clients have sparse label distributions, highlighting the need for target-aware aggregation strategies.

## 2   Target-aware federated learning with label shift

In federated learning, a global model $h_\theta$ is produced by a central server by aggregating updates to model parameters $\theta$ from multiple clients (McMahan et al., 2017). Here, we focus on classification tasks in which the goal is for $h_\theta$ to predict the most probable label $Y \in \{1, ..., K\}$ for a given $d$-dimensional input $X \in \mathcal{X} \subset \mathbb{R}^d$. Each client $i = 1, ..., M$ holds a data set $D_i = \{(x_{i,1}, y_{i,1}), ..., (x_{i,n_i}, y_{i,n_i})\}$ of $n_i$ labeled examples, assumed to be drawn i.i.d. from a *local* client-specific distribution $S_i(X, Y)$. Due to constraints on privacy or communication, these data sets cannot be shared directly with other clients or with the server.

Learning proceeds over rounds $t = 1, ..., t_{max}$, each comprising three steps: (1) The central server broadcasts the current global model parameters $\theta_t$ to all clients; (2) Each client $i$ computes updated parameters $\theta_{i,t}$ based on their local data set $D_i$, and sends these updates back to the server; (3) The server aggregates the clients' updates, for example, using federated averaging (FedAvg) (McMahan et al., 2017) or related techniques, to produce the new global model $\theta_{t+1}$.

A common implicit assumption in federated learning is that the learned model will be applied in a target domain $T(X, Y)$ that coincides with the aggregate distribution of clients,[1]

$$\bar{S}(X, Y) = \sum_{i=1}^{M} \frac{n_i}{N} S_i(X, Y), \tag{1}$$

where $N = \sum_{i=1}^{M} n_i$. Consequently, trained models are evaluated in terms of their average performance over clients. However, this aggregate depends both on the number of samples each client possesses and the set of included clients itself, both of which may change over time. In some applications, the intended target domain may be entirely different from the client training sets by design (Bai et al., 2024). Distributional

---

[1]Personalized federated learning (Fallah et al., 2020) is an important exception, but also breaks with the goal of producing a single global model. We do not consider this setting further.

shift *between clients* is a well-recognized problem in federated learning, the target domain is still typically the client aggregate $\bar{S}$ (Karimireddy et al., 2020; Li et al., 2020b).

We consider a generalization of the standard setting where a model is trained to perform well in a target domain $T(X, Y)$, without access to samples from it. Our objective is to minimize the expected target risk, $R_T$ of a classifier $h_\theta : \mathcal{X} \to \mathcal{Y}$, with respect to a loss function $\ell : \mathcal{Y} \times \mathcal{Y} \to \mathbb{R}$,

$$\underset{\theta}{\text{minimize }} R_T(h_\theta) := \underset{(X,Y) \sim T}{\mathbb{E}} [\ell(h_\theta(X), Y)] . \tag{2}$$

The target domain $T(X, Y)$ is free to be distinct from any client distributions, $\forall i : T(X, Y) \neq S_i(X, Y)$, and from the client aggregate, $T(X, Y) \neq \bar{S}(X, Y)$. This differs from federated domain generalization which lacks a specific target domain (Bai et al., 2024). Two central questions become: How sensitive are existing federated learning algorithms to the choice of target domain, and can we do better if we know something about $T$? We refer to the latter as *target-aware federated learning.*

To implement target-aware federated learning, we assume that target and client label marginal distributions $T(Y), \{S_i(Y)\}$ are known to the central server. As we will see empirically, giving the server such access is well justified by consistent performance gains across tasks. However, it also raises two potential tradeoffs, (a) statistics and (b) privacy. First, estimating each client label distribution $S_i(Y)$ merely involves computing the proportion of each label in the client sample $D_i$. This comes at minimal cost. For the target domain, the federation (or target client) may have collected label statistics without logging context features $X$. In our retail example, the label distribution corresponds to the proportion of sales $T(Y = y)$ of each product category $y$. Many companies store this information without logging customer features $X$. If the target domain includes a new company, their label distribution can be readily incorporated.

When it comes to privacy, the central server is given access to all label distributions to facilitate the learning process, but these are *not available to the clients.* Retailers may be hesitant to share their exact sales statistics $T(Y)$ with competitors but could share this information with a neutral third party (central server) responsible for coordinating the federated learning process. Moreover, the marginal label distribution is typically much less sensitive than the data itself. There is a privacy-accuracy trade-off in all FL settings, and it is important to understand what are the potential performance gains for raised privacy costs. In our experiments, *we show that substantial performance improvements can be gained at the small cost of sharing the label marginals with the central server.*

As in standard FL, clients $i \neq j$ do not communicate with each other directly but interact with the central server through model parameters. While it is technically possible for the server to *infer* each client's label distribution $S_i(Y)$ based on their parameter updates (Ramakrishna & Dán, 2022), doing so would likely be considered a breach of trust in practical applications and sharing would be preferred.

We assume that the distributional shifts between clients and the target are restricted to *label shift*—while the label distributions vary across clients and the target, the class-conditional input distributions are identical. **Assumption 1** (Label shift)**.** *For the client distributions $S_1, ..., S_M$ and the target distribution $T$,*

$$\forall i, j \in [M] : S_i(X \mid Y) = S_j(X \mid Y) = T(X \mid Y) . \tag{3}$$

This setting has been well studied in non-federated learning, see e.g., (Lipton et al., 2018). In the retail example, label shift means that the proportion of sales across product categories ($S_i(Y)$ and $T(Y)$) varies between different retailers and the target, but that the pattern of customers who purchase items in each category ($S_i(X \mid Y)$ and $T(X \mid Y)$) remain consistent. In other words, although retailers may sell different quantities of products across categories, the characteristics of customers buying a particular product (conditional on the product category) are assumed to be the same. Note that both label shift and *covariate shift* may hold, that is, there are cases where $\forall i : S_i(X \mid Y) = T(X \mid Y)$ *and* $S_i(Y \mid X) = T(Y \mid X)$, but $S_i(X), T(X)$ differ, such as when the labeling function is deterministic.

## 2.1 Limitations of classical aggregation

When either all clients $\{S_i\}$ or their aggregate $\bar{S}$, see equation 1, are identical in distribution to the target domain $T$, the empirical risk $\hat{R}$ on aggregated client data is identical in distribution ($\overset{d}{=}$) to the empirical risk

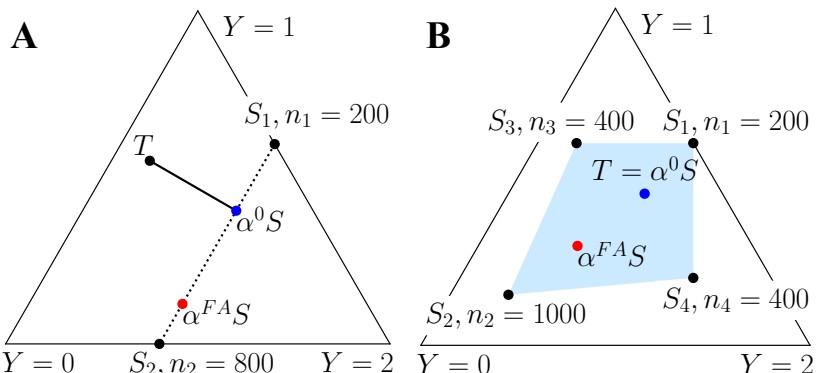

Figure 1: Illustration of the target label marginal $T$ and client marginals $S_1, ..., S_4$ in a ternary classification task, $Y \in \{0, 1, 2\}$. A: there are fewer clients than labels, $M < K$, and $T \notin \text{Conv}(S)$; $\alpha^0 S$ is a projection of $T$ onto $\text{Conv}(S)$. B: $T \in \text{Conv}(S)$ and coincides with $\alpha^0 S$. In both cases, the label marginal $\alpha^{FA} S$ implied by FedAvg is further from the target distribution.

$\hat{R}_T$ of a hypothetical data set $D_T = \{(x_{T,j}, y_{T,j})\}_{j=1}^{n_T}$ drawn from the target domain,

$$\hat{R} := \sum_{i=1}^{M} \sum_{j=1}^{n_i} \frac{\ell(h(x_{i,j}), y_{i,j})}{N} \stackrel{d}{=} \sum_{j=1}^{n_T} \frac{\ell(h(x_{T,j}), y_{T,j})}{n_T} =: \hat{R}_T .$$

As a consequence, if each client performs a single gradient descent update, the mean of these, weighted by the client sample sizes, is equal in distribution to a centralized batch update for the target domain, given the previous parameter value. This property justifies the federated stochastic gradient (FedSGD) and federated averaging principles (McMahan et al., 2017), both of which aggregate parameter updates in this way,

$$\theta_{t+1}^{FA} = \sum_{i=1}^{M} \alpha_i^{FA} \theta_{i,t} \quad \text{where} \quad \alpha_i^{FA} = \frac{n_i}{\sum_{j=1}^{M} n_j} . \tag{4}$$

When the target domain $T$ is *not* the aggregate of clients $\bar{S}$, the aggregate risk gradient $\nabla_\theta \hat{R}$ and, therefore, the FedSGD update are no longer unbiased gradients and updates for the risk in the target domain. As we see in experiment results in Table 1, this can have large effects on model quality. The fact that clients are ignorant of the shift in $T(Y)$ affects the optimal strategy; direct access to $T(Y)$ would allow sample re-weighting or upsampling in the client objectives (Rubinstein & Kroese, 2016). This is not possible here.

**Our central question is:** How can we *aggregate* the parameter updates $\theta_{i,t}$ of the clients, whose data sets are drawn from distributions $S_1, ..., S_M$, such that the resulting federated learning algorithm minimizes $R_T$?

## 3 FedPALS: Target-aware adjustment for label shift

Next, we develop a model aggregation strategy for target-aware federated learning. Under Assumption 1 (label shift), it is easy to show that the target risk is a weighted sum of class-conditional client risks,

$$\forall i : R_T(h) = \sum_{y=1}^{K} T(y) \mathbb{E}_{S_i}[\ell(h(X), y) \mid Y = y] ,$$

provided that each client has support for each label, that is $S_i(X \mid Y = y)$ exists for all $y$. In centralized learning, this insight is often used to re-weight the training objective in a source domain $S(y)$ by the importance ratio $T(y)/S(y)$ (Lipton et al., 2018; Japkowicz & Stephen, 2002). *However, that is not an option here since $T(Y)$ is not revealed to the clients.* Thus, only the server can incorporate knowledge of the label importance ratio for target-aware learning.

In practice, assuming that each label $y$ is supported by each client $i$, $S_i(Y = y) > 0$ is often too strong. In fact, many of the widely used benchmarks for federated learning violate this assumption (Koh et al., 2021). Another interesting case is when the target label distribution lies in the convex hull of the set of client label distributions $S = \{S_i(Y)\}_{i=1}^M$—that it can be recovered as a convex combination of clients (Figure 1 B).

**Assumption 2** (Target coverage). *The target label distribution $T(Y)$ is covered by the convex hull of client label distributions $S_1(Y), ..., S_M(Y)$, that is $T \in \text{Conv}(S)$, or*

$$\exists \omega \in \Delta^{M-1} : T(y) = \sum_{i=1}^M \omega_i S_i(y) \quad \forall y \in [K] \; . \tag{5}$$

Under label shift, target coverage (Assumption 2) implies that $T(X, Y) = \sum_{i=1}^M \omega_i S_i(X, Y)$ for some convex combination $\omega$. Thus, under Assumptions 1–2, we have for any $\omega$ satisfying equation 5,

$$R_T(h) = \sum_{y=1}^K \left( \sum_{i=1}^M \omega_i S_i(y) \right) \mathbb{E}[\ell(h(X), y) \mid Y = y] \tag{6}$$

$$= \sum_{i=1}^M \omega_i R_{S_i}(h) \; .$$

Consequently, aggregating client gradients with weights $\omega$ yields an unbiased estimate of the gradient descent update for the target risk in federated stochastic gradient descent.

**Proposition 1** (Unbiased SGD update). *Consider a single round $t$ of federated learning in the batch stochastic gradient setting with learning rate $\eta$. Each client $i \in [M]$ is given parameters $\theta_t$ by the server, computes their local gradient, and returns the update $\theta_{i,t} = \theta_t - \eta \nabla_\theta \hat{R}_i(h_{\theta_t})$. Let Assumptions 1–2 hold and $\omega$ satisfy equation 5. Then, the aggregate update $\theta_{t+1} = \sum_{i=1}^M \omega_i \theta_{i,t}$ satisfies*

$$\mathbb{E}[\theta_{t+1} \mid \theta_t] = \mathbb{E}[\theta_{t+1}^T \mid \theta_t] \; ,$$

*where $\theta_{t+1}^T = \theta_t - \eta \nabla_\theta \hat{R}_T(h_{\theta_t})$ is the batch stochastic gradient descent (SGD) update for $\hat{R}_T$ that would be obtained with a sample from the target domain. A proof is in givenAppendix D.*

By Proposition 1, we may compute unbiased parameter updates for the target domain by replacing the aggregation step of FedSGD with aggregation weighted according to $\omega$. In practice, many federated learning systems, including FedAvg, allow clients *several steps* of local optimization (e.g., an epoch) before aggregating the parameter updates at the server. Strictly speaking, this is not justified by Proposition 1, but we find in all experiments that aggregating client updates computed over an epoch performs well, see Section 5.

### 3.1  Adjustment without target coverage

In applications, Assumption 2 may not hold—the target may not be covered by clients—and $\omega$ in equation 5 may not exist. For example, if the target label marginal $T(y)$ is sparse, only clients with *at least the same unsupported labels* as $T$ can be used in a convex combination $\omega S = T$. That is, for any client $S_i$ with $\omega_i > 0$, it must hold for any $y$ such that $T(y) = 0$ that $S_i(y) = 0$ as well; otherwise, including the client would introduce support on a label unsupported in the target. For example, if we aim to classify images of animals and $T$ contains no tigers, then no clients contributing to the combination can have data containing tigers. Since $\{S_i(Y)\}_{i=1}^M, T(Y)$ are known to the server, it is straightforward to verify Assumption 2.

A pragmatic choice when Assumption 2 is violated is to look for the convex combination $\alpha^0$ that most closely aligns with the target label distribution, and use that for aggregation,

$$\alpha^0 = \operatorname*{arg\,min}_{\alpha \in \Delta^{M-1}} \left\| \sum_{i=1}^M \alpha_i S_i(Y) - T(Y) \right\|_2^2 \tag{7}$$

We illustrate the label distributions implied by weighting with $\alpha^0$ and $\alpha^{FA}$ (FedAvg) in Figure 1. Unlike $\omega$, $\alpha^0$ always exists and can be used for weighted aggregation.

**Effective sample size of aggregates.** A limitation of aggregating client updates using $\alpha^0$ as defined in equation 7 is that, unlike other aggregation methods, it does not give higher weight to clients with larger sample sizes. This can lead to a higher variance in the model estimate as it may prioritize clients with very small training sets. The variance of importance-weighted estimators can be quantified through the *effective sample size* (ESS) (Kong, 1992), which measures the number of samples needed from the target domain to achieve the same variance as a weighted estimate computed from source-domain samples. ESS is often approximated as $1/(\sum_{i=1}^m w_j^2)$ where $w$ are normalized sample weights such that $w_j \geq 0$ and $\sum_{j=1}^n w_j = 1$.

In federated learning, we can interpret the aggregation step (e.g., equation 1) as assigning a total weight $\alpha_i$ to a client $i$, which has $n_i$ samples. Each sample $(x_j, y_j) \in D_i$ in the client's training set is given the same weight $\tilde{w}_j = \alpha_i / n_i$. The effective sample size for the weighted aggregate is then given by $1/(\sum_{i=1}^m (\sum_{j \in S_i} \tilde{w}_j^2)) = 1/(\sum_{i=1}^m n_i \alpha_i^2 / n_i^2) = 1/(\sum_{i=1}^m \alpha_i^2 / n_i)$ which grows when large-sample clients are given high relative weight.

In light of the above, we propose a client aggregation step such that the weighted sum of clients' label distributions will a) closely align with the target label distribution, and b) minimize the variance due to weighting, by regularizing the projection onto the convex hull using the inverse of the ESS. For a given regularization parameter $\lambda \in [0, \infty)$, we define weights $\alpha^\lambda$ as the solution to the following problem

$$\alpha^\lambda = \arg\min_{\alpha \in \Delta^{M-1}} \|T(Y) - \sum_{i=1}^M \alpha_i S_i(Y)\|_2^2 + \lambda \sum_i \frac{\alpha_i^2}{n_i} \ , \tag{8}$$

with aggregate client parameters as $\theta_{t+1}^\lambda = \sum_{i=1}^M \alpha_i^\lambda \theta_{i,t}$. We refer to this strategy as Federated learning with Parameter Aggregation for Label Shift (FedPALS). Setting $\lambda = 0$, we recover $\alpha^0$ from before.

**Computational cost.** The problem in equation 8 is a quadratic program with a simplex constraint, which can be solved readily using off-the-shelf solvers. General-purpose algorithms, such as interior-point methods, take $O(M^3)$ time in the worst case, but the time to solve it is negligible in practice, usually on the order of seconds. Moreover, the combination $\alpha^\lambda$ only has to be computed *once*, before federated learning starts. Similar to many other machine learning strategies, the FedPALS weight-learning equation 8 objective includes a hyperparameter, $\lambda$. For optimal performance, $\lambda$ should be treated as a hyperparameter of the entire federated learning process and could be selected by training multiple models with the federation, for multiple values of $\lambda$, and selecting the best model on a held-out validation set. This is not uncommon in federated learning strategies. If re-training models is too expensive, $\lambda$ may be set heuristically to yield a given effective sample size, see Appendix B.3. For larger training sets, a smaller ESS can be tolerated.

## 3.2 FedPALS in the limits

In the FedPALS aggregation scheme, equation 8, there is a trade-off between closely matching the target label distribution and minimizing the variance of the model parameters. This trade-off exhibits some interesting limit cases. In particular, in the limit $\lambda \to \infty$, as the regularization parameter $\lambda$ grows large, FedPALS aggregation approaches standard FedAvg aggregation.

**Proposition 2.** *The limit solution $\alpha^\lambda$ to equation 8, as $\lambda \to \infty$, is*

$$\lim_{\lambda \to \infty} \alpha_i^\lambda = \frac{n_i}{\sum_{j=1}^M n_j} = \alpha_i^{FA} \quad for \quad i = 1, \ldots, M \ . \tag{9}$$

The result is proven in Appendix D. By Proposition 2, the FedAvg weights $\alpha^{FA}$ minimize the inverse ESS and coincide with FedPALS weights $\alpha^\lambda$ in the limit $\lambda \to \infty$. As a rare special case, whenever the target is equal to the client aggregate, $T(Y) = \bar{S} = \sum_{i=1}^M \frac{n_i}{N} S_i(Y)$, FedAvg weights equal FedPALS weights, $\alpha^{FA} = \alpha^\lambda$, for any value of $\lambda$, since both terms attain their mimima at this point. However, this violates the assumption that $T(Y) \neq \bar{S}(Y)$. We discuss some other cases, including $T \in \text{Conv}(S), \lambda \to 0$, in Appendix B.2.

**Sparse clients and targets** In problems with a large number of labels, $K \gg 1$, it is common that any individual domain (clients or target) supports only a subset of the labels. For example, in the widely-used

IWildCam benchmark (Koh et al., 2021), not every wildlife camera captures images of all animal species. When the target $T(Y)$ is *sparse*, meaning $T(y) = 0$ for certain labels $y$, it becomes easier to find a good match $(\alpha^\lambda)^\top S(Y) \approx T(Y)$ if the client label distributions are also sparse. Achieving a perfect match, i.e., $T \in \mathrm{Conv}(S)$, requires that (i) the clients collectively cover all labels in the target, and (ii) each client contains only labels that are present in the target. If this is also beneficial for learning, it would suggest, rather unintuitively, that the client-presence of labels that are not present in the target would *harm* the aggregated model. We study the implications of sparsity of label distributions empirically in Section 5.

## 4 Related work

Efforts to mitigate the effects of distributional shifts in federated learning can be broadly categorized into client-side and server-side approaches. Client-side methods use techniques such as clustering clients with similar data distributions and training separate models for each cluster (Ghosh et al., 2020; Sattler et al., 2020; Vardhan et al., 2024), and meta-learning to enable models to quickly adapt to new data distributions with minimal updates (Chen et al., 2018; Jiang et al., 2019; Fallah et al., 2020). Other notable strategies include regularization techniques that penalize large deviations in client updates to ensure stable convergence (Li et al., 2020b; 2021) and recent work on optimizing for flatter minima to enhance model robustness (Qu et al., 2022; Caldarola et al., 2022). Server-side methods focus on improving model aggregation or applying post-aggregation adjustments (Breitholtz et al., 2025). These include optimizing aggregation weights (Reddi et al., 2021), learning adaptive weights (Li et al., 2023), iterative moving averages to refine the global model (Zhou et al., 2023), and promoting gradient diversity during updates (Zeng et al., 2023). Both categories of work overlook shifts in the target distribution, leaving this area unexplored.

Another related area is personalized federated learning, which focuses on fine-tuning models to optimize performance on each client's specific local data (Collins et al., 2022; Boroujeni et al., 2024). This setting differs fundamentally from our work, which focuses on improving generalization to new target clients without any training data available for fine-tuning. Label distribution shifts have also been explored with methods such as logit calibration (Zhang et al., 2022; Xu et al., 2023), novel loss functions (Wang et al., 2021), feature augmentation (Xia et al., 2023), gradient reweighting (Xiao et al., 2023), and contrastive learning (Wu et al., 2023). However, like methods aimed at mitigating the effects of general shifts, these do not address the challenge of aligning models with an unseen target distribution, as required in our setting.

Generalization under domain shift in federated learning remains underdeveloped (Bai et al., 2024). The work most similar to ours is that of agnostic federated learning (AFL) (Mohri et al., 2019), which aims to learn a model that performs robustly across *all* possible target distributions within the convex hull of client distributions. One notable approach is tailored for medical image segmentation, where clients share data in the frequency domain to achieve better generalization across domains (Liu et al., 2021). However, this technique requires data sharing, making it unsuitable for privacy-sensitive applications like ours. A different line of work focuses on addressing covariate shift in federated learning through importance weighting (Ramezani-Kebrya et al., 2023). Although effective, this method requires sending samples from the test distribution to the server, which violates our privacy constraints.

## 5 Experiments

We perform a series of experiments on diverse tasks, with $M \in \{3, 10, 100\}$ clients, built from established benchmark data sets to evaluate FedPALS in comparison with baseline federated learning algorithms. The experiments aim to demonstrate the value of the central server knowing the label distributions of the client and target domains when these differ substantially. Additionally, we seek to understand how the parameter $\lambda$, controlling the trade-off between bias and variance in the FedPALS aggregation scheme, impacts the results. Finally, we investigate how the benefits of FedPALS are affected by the sparsity of label distributions and by the distance $d(T, S) \coloneqq \min_{\alpha \in \Delta^{M-1}} \|T(Y) - \alpha^\top S(Y)\|_2^2$ from the target to the convex hull of clients.

**Experimental setup** While numerous benchmarks exist for federated learning (Caldas et al., 2018; Chen et al., 2022) and domain generalization (Gulrajani & Lopez-Paz, 2020; Koh et al., 2021), respectively, until

Table 1: Comparison of mean accuracy and standard deviation ($\pm$) across different algorithms. The reported values are over 8 independent random seeds for the CIFAR-10 and Fashion-MNIST tasks, and 3 for PACS. $C$ indicates the number of labels per client and $\beta$ the Dirichlet concentration parameter. $M$ is the number of clients. The *Oracle* method refers to a FedAvg model trained on client distributions identical to the target.

| Data set | Label split | M | FedPALS | FedAvg | FedProx | SCAFFOLD | AFL | FedRS | Oracle |
|---|---|---|---|---|---|---|---|---|---|
| Fashion-MNIST | $C = 3$ | 10 | $\mathbf{92.4 \pm 2.1}$ | $67.1 \pm 22.0$ | $66.9 \pm 20.8$ | $69.5 \pm 19.3$ | $78.9 \pm 14.7$ | $85.3 \pm 13.5$ | $97.6 \pm 2.1$ |
|  | $C = 2$ |  | $\mathbf{80.6 \pm 23.7}$ | $53.9 \pm 36.2$ | $52.9 \pm 35.7$ | $54.9 \pm 36.8$ | $78.6 \pm 20.0$ | $63.14 \pm 20.2$ | $97.5 \pm 4.0$ |
| CIFAR-10 | $C = 3$ |  | $\mathbf{65.6 \pm 10.1}$ | $44.0 \pm 8.4$ | $43.5 \pm 7.2$ | $43.3 \pm 7.4$ | $53.2 \pm 0.9$ | $44.0 \pm 8.0$ | $85.5 \pm 5.0$ |
|  | $C = 2$ | 10 | $\mathbf{72.8 \pm 17.4}$ | $46.7 \pm 15.8$ | $47.7 \pm 15.6$ | $46.7 \pm 14.9$ | $54.7 \pm 0.1$ | $49.4 \pm 9.5$ | $89.2 \pm 3.9$ |
|  | $\beta = 0.1$ |  | $\mathbf{62.6 \pm 17.9}$ | $40.8 \pm 9.2$ | $41.9 \pm 9.7$ | $43.5 \pm 10.5$ | $53.4 \pm 11.5$ | $57.1 \pm 11.2$ | $79.2 \pm 3.7$ |
| PACS | $C = 6$ | 3 | $\mathbf{86.0 \pm 2.9}$ | $73.4 \pm 1.6$ | $75.3 \pm 1.3$ | $73.9 \pm 0.3$ | $74.5 \pm 0.9$ | $76.1 \pm 1.6$ | $90.5 \pm 0.3$ |

recently none have addressed tasks that combine both settings. To fill this gap, Bai et al. (2024) introduced a benchmark specifically designed for federated domain generalization (DG), evaluating methods across diverse datasets with varying levels of client heterogeneity. In our experiments, we use the PACS Li et al. (2017) and iWildCAM data sets from the Bai et al. (2024) benchmark to model realistic label shifts between the client and target distributions. We modify the PACS dataset to consist of three clients, each missing a label that is present in the other two. Additionally, one client is reduced to one-tenth the size of the others, and the target distribution is made sparse in the same label as that of the smaller client. PACS ($M = 3$) represents a few-client scenario, and iWildCam ($M = 100$) a large-scale many-client task. See Appendix B for details.

Furthermore, we construct two additional tasks by introducing label shift to standard image classification data sets, Fashion-MNIST (Xiao et al., 2017) and CIFAR-10 (Krizhevsky, 2009). We apply two label shift sampling strategies: sparsity sampling and Dirichlet sampling. Sparsity sampling involves randomly removing a subset of labels from clients and the target domain, following the data set partitioning technique first introduced in McMahan et al. (2017). Each client is assigned $C$ random labels, with an equal number of samples for each label and no overlap among clients. Dirichlet sampling simulates realistic non-i.i.d. label distributions by, for each client $i$, drawing a sample $p_i \sim \text{Dirichlet}_K(\beta)$, where $p_i(k)$ represents the proportion of samples in client $i$ that have label $k \in [K]$. We use a symmetric concentration parameter $\beta > 0$ which controls the sparsity of the client distributions. See Appendix B.4.

While prior works have focused on inter-client distribution shifts assuming that client and target domains are equally distributed, *we apply these sampling strategies also to the target set*, thereby introducing label shift between the client and target data. Figures 2b & 5b (latter in appendix) illustrate an example with $C = 6$ for sparsity sampling and Dirichlet sampling with $\beta = 0.1$, where the last client (Client 9) is chosen as the target. In addition, we investigate the effect of $T(Y) \notin \text{Conv}(S)$ in a task described in C.4.

**Baseline algorithms and model architectures**   Alongside FedAvg, we use SCAFFOLD, FedProx, AFL and FedRS (Karimireddy et al., 2020; Li et al., 2020b; Mohri et al., 2019; Li & Zhan, 2021) as baselines. The first two chosen due to their prominence in the literature for handling non-iid data, and AFL which is similar in concept to FedPALS and aims to optimize for an unseen domain. We also include FedRS, designed specifically to address label distribution skew. For the synthetic experiment in Section C.4, we use a logistic regression model. For CIFAR-10 and Fashion-MNIST, we use small, two-layer convolutional networks, while for PACS and iWildCAM, we use a ResNet-50 pre-trained on ImageNet. Early stopping, model hyperparameters, and $\lambda$ in FedPALS are tuned using a validation set that reflects the target distribution in the synthetic experiment, CIFAR-10, Fashion-MNIST, and PACS. This tuning process consistently resulted in setting the number of local epochs to $E = 1$ across all experiments. For iWildCAM, we adopt the hyperparameters reported by Bai et al. (2024) and select $\lambda$ using the same validation set used in their work. We report the mean test accuracy and standard deviation for each method over 3 independent random seeds for PACS and iWildCam and 8 seeds for the Fashion-MNIST and CIFAR-10, to ensure robust evaluation.

## 5.1 Experimental results on benchmark tasks

We present results for three tasks with selected skews in Table 1 and explore detailed results below. Across these tasks, FedPALS consistently outperforms or matches the best-performing baseline. For PACS, Fashion-

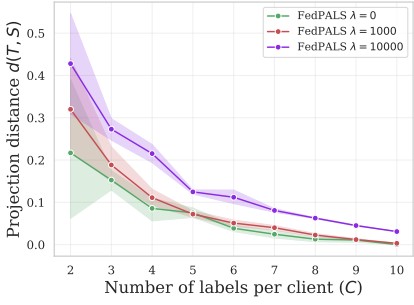
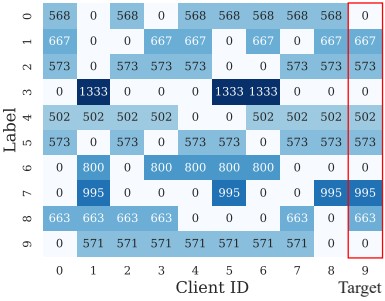
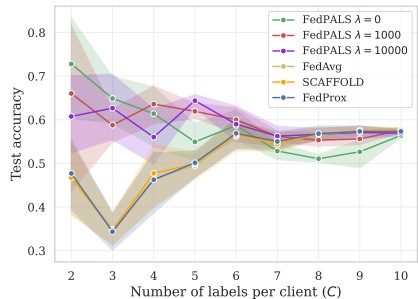

(a) Projection distance between target and client convex hull, varying $C$.

(b) Marginal label distributions of clients and target with $C = 6$ labels per client.

(c) Accuracy vs labels per client, $C$.

Figure 2: Results on CIFAR-10 with sparsity sampling, varying the number of labels per clients $C$ across 10 clients. Clients with IDs 0–8 are used in training, and Client 9 is the target client. The task is more difficult for small $C$, when fewer clients share labels, and the projection distance is larger. Note that the target label distribution also becomes sparser when $C$ is reduced.

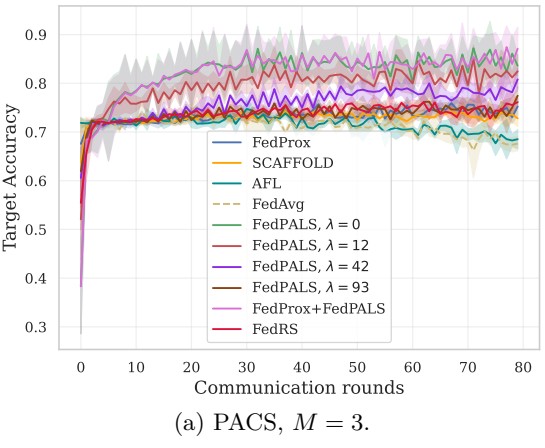
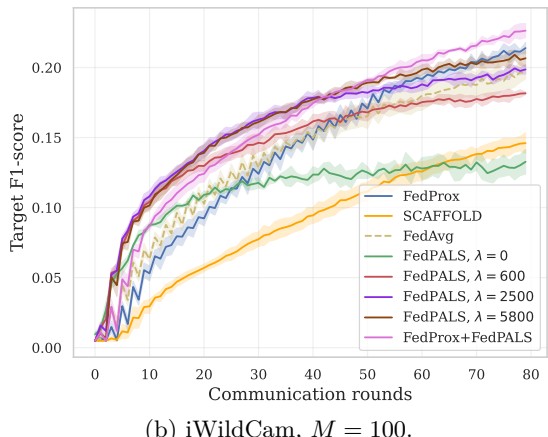

(a) PACS, $M = 3$.

(b) iWildCam, $M = 100$.

Figure 3: Target accuracy/F1-score during training of FedPALS compared to baselines on PACS (a) and iWildCam (b), averaged over 3 random seeds. $M$ is the number of training clients. Non-zero $\lambda$-values chosen to correspond to an ESS of 25%, 50% and 75%.

MNIST and CIFAR-10, we include results for an *Oracle* FedAvg model, which is trained on clients whose distributions are identical to the target distribution, eliminating any client-target distribution shift (see Appendix B for details). A FedPALS-*Oracle* would be equivalent since there is no label shift. The *Oracle*, which has perfect alignment between client and target distributions, achieves superior performance, underscoring the challenge posed by distribution shifts in real-world scenarios where such alignment is absent.

**CIFAR-10/Fashion-MNIST.** Figure 2c shows the results for the CIFAR-10 data set, where we vary the label sparsity across clients. In the standard i.i.d. setting, where all labels are present in both the training and target clients ($C = 10$), all methods perform comparably. As label sparsity increases and fewer labels are available in client data sets (i.e., as $C$ decreases), we observe a performance degradation in standard baselines. In contrast, our proposed method, FedPALS, leverages optimized aggregation to achieve a lower target risk, resulting in improved test accuracy under these challenging conditions. Similar trends are observed for Fashion-MNIST, as shown in Figure 6 in Appendix C. Furthermore, the results in the highly non-i.i.d. cases ($C = 2, 3$ and $\beta = 0.1$) are summarized in Table 1. Additional experiments in Appendix C examine how the algorithms perform with varying numbers of local epochs (up to 40) and clients (up to 100).

**PACS.** As shown in Figure 3a, being faithful to the target distribution is crucial for improved performance. Lower values of $\lambda$ generally correspond to better performance. Notably, FedAvg struggles in this setting because it systematically underweights the client with the distribution most similar to the target, leading to

suboptimal model performance. In fact, this even causes performance to degrade over time. Interestingly, the baselines also face challenges on this task: both FedProx, FedRS and SCAFFOLD perform similarly to FedPALS when $\lambda = 93$. However, FedPALS demonstrates significant improvements over these methods, highlighting the effectiveness of our aggregation scheme in enhancing performance. We also see that FedPALS + FedProx performs comparably to just using FedPALS in this case, although it does have higher variance. Additionally, in Table 1, we present the models selected based on the source validation set, where FedPALS outperforms all other methods. For comprehensive results, including all FedPALS models and baseline comparisons, see Table 4 in Appendix C.

**iWildCam.** The test performance across communication rounds is shown in Figure 3b. Initially, FedPALS widens the performance gap compared to FedAvg, but as training progresses, this gain diminishes. While FedPALS quickly reaches a strong performing model, it eventually plateaus. The rate of convergence and level of performance reached appears to be influenced by the choice of $\lambda$, with lower values of $\lambda$ leading to faster plateaus at lower levels compared to larger ones. This suggests that more uniform client weights and a larger effective sample size are preferable in this task. Given the iWildCam dataset's significant class imbalance – with many classes having few samples – de-emphasizing certain clients can degrade performance. We also note that our assumption of label shift need not hold in this experiment, as the cameras are in different locations, potentially leading to variations in the conditional distribution $p(X \mid Y)$. The performance of the models selected using the source validation set is shown in Table 3 in Appendix C. There we see that FedPALS performs comparably to FedAvg and FedProx while outperforming SCAFFOLD. Unlike in other tasks, where FedProx performs comparably or worse than FedPALS, we see FedProx achieve the highest F1-score on this task. Therefore, we conduct an additional experiment where we use both FedProx and FedPALS together, as they are not mutually exclusive. This results in the best performing model, see Figure 3b. Due to memory issues with the implementation FedRS was not able to run for this experiment and is omitted. AFL fails to learn in this task and is thus also omitted, although results are shown in Table 3 in Appendix C. Finally, as an illustration of the impact of increasing $\lambda$, we provide the weights of the clients in this experiment alongside the FedAvg weights in Figure 4 in Appendix C. We note that as $\lambda$ increases, the weights increasingly align with those of FedAvg while retaining weight on the clients whose label distributions most resemble that of the target.

## 6 Discussion

We have explored *target-aware federated learning under label shift*, a scenario where client data distributions differ from a target domain with a known label distribution, but no target samples are available. We demonstrated that traditional approaches, such as federated averaging (FedAvg), which assume identical distributions between the client aggregate and the target, fail to adapt effectively in this context due to biased aggregation of client updates. To address this, we proposed FedPALS, a novel aggregation strategy that optimally combines client updates to align with the target distribution, ensuring that the aggregated model minimizes target risk. Empirically, across diverse tasks, we showed that under label shift, FedPALS substantially outperforms standard methods like FedAvg, FedProx, FedRS and SCAFFOLD, as well as AFL. Specifically, when the target label distribution lies within the convex hull of the client distributions, FedPALS finds the solution with the largest effective sample size, leading to a model that is most faithful to the target distribution. More generally, FedPALS balances the trade-off between matching the target label distribution and minimizing variance in the model updates. Our experiments further highlight that FedPALS excels in challenging scenarios where label sparsity and client heterogeneity hinder the performance of conventional federated learning methods.

One of the limitations of FedPALS is the reliance on label(-only) shift—the assumption that label-conditional input distributions are equal in all clients and the target. We observed empirically that selecting the trade-off parameter $\lambda$ is crucial for optimal performance in tasks such as iWildCam, where this assumption may not fully hold. Future work should aim to detect and overcome violations of this assumption. Moreover, FedPALS can underperform in scenarios where one or more clients, which are essential for accurately mirroring the target distribution, have limited sample sizes, and $\lambda$ is set too low. In such cases, the effective sample size of the aggregated dataset becomes insufficient, potentially hindering the model's ability to learn effectively.

Further, when the client aggregate is identical to the target, we do not expect FedPALS to produce better solutions than standard aggregation using FedAvg as the methods are equivalent in this case.

Similar to many methods in FL, there is an inherent privacy-accuracy trade-off to the target-aware federated learning setting. Here, we have demonstrated that substantial improvements in accuracy can be achieved at the cost of clients sharing their label marginals with the central server. Further analyzing the extent to which partially obfuscated label marginals can be shared and used to improve target-awareness is an interesting direction for future work. Another generalization would be to support federated regression problems, which have a fundamentally different structure to multi-class classification tasks. Due to the continuous distribution of regression labels, adjusting for label shift requires a substantially different machinery from what we propose here, but is feasible using, e.g., kernel density estimation. Finally, the early performance gains observed during training suggest that tuning the regularization parameter $\lambda$ during training could enhance performance of FedPALS. A promising avenue for future work would be exploring adaptive strategies for dynamically tuning $\lambda$.

### Acknowledgments

This work was supported in part by the Wallenberg AI, Autonomous Systems and Software Program (WASP) funded by the Knut and Alice Wallenberg Foundation. The computations and data handling were enabled by resources provided by the National Academic Infrastructure for Supercomputing in Sweden (NAISS), partially funded by the Swedish Research Council through grant agreements no. 2022-06725 and 2024-03903.

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

## A    Appendix

## B    Experimental details

Here we provide additional details about the experimental setup for the different tasks. The code will be made available upon acceptance.

### B.1    Algorithm

---
**Algorithm 1** FedPALS

---
**Require:** Client label distributions $\{\mathcal{S}_i(Y)\}$, target label distribution $\mathcal{T}(Y)$, trade-off parameter $\lambda$
**Ensure:** Classifier $h(x)$

1: Initialize central parameters $\theta_0$
2: Solve $\alpha^\lambda = \arg\min_{\alpha \in \Delta^{M-1}} \left\| \mathcal{T}(Y) - \sum_{i=1}^{M} \alpha_i \mathcal{S}_i(Y) \right\|_2^2 + \lambda \sum_i \frac{\alpha_i^2}{n_i}$
3: **for** round $t = 0, \ldots, T-1$ **do**
4:     **for** each client $k = 1, \ldots, m$ **do**
5:         Distribute $\theta_t$ to client $k$
6:         Receive client update $\theta_{k,t}$
7:     **end for**
8:     $\theta_{t+1} = \sum_{i=1}^{m} \alpha_i^\lambda \theta_{i,t}$
9: **end for**
10: **return** classifier with parameters $\theta_{t+1}$

---

### B.2    FedPALS in the limit

There are some interesting limit cases of FedPALS which we discuss here. Notably, in the first case, FedPALS is shown to be equivalent to FedAvg.

**Case 1:** $\lambda \to \infty \Rightarrow$ **Federated averaging**    In the limit $\lambda \to \infty$, as the regularization parameter $\lambda$ grows large, FedPALS aggregation approaches FedAvg aggregation.

**Proposition 3.** *The limit solution $\alpha^\lambda$ to equation 8, as $\lambda \to \infty$, is*

$$\lim_{\lambda \to \infty} \alpha_i^\lambda = \frac{n_i}{\sum_{j=1}^{M} n_j} = \alpha_i^{FA} \quad for \quad i = 1, \ldots, M \ . \tag{10}$$

The result is proven in Appendix D. By Proposition 2, the FedAvg weights $\alpha^{FA}$ minimize the ESS and coincide with FedPALS weights $\alpha^\lambda$ in the limit $\lambda \to \infty$. As a rare special case, whenever $T(Y) = \bar{S} = \sum_{i=1}^{M} \frac{n_i}{N} S_i(Y)$, FedAvg weights $\alpha^{FA} = \alpha^\lambda$ for any value of $\lambda$, since both terms attain their mimima at this point. However, this violates the assumption that $T(Y) \neq \bar{S}(Y)$.

**Case 2:  Covered target,** $T \in \mathrm{Conv}(S)$    Now, consider when the target label distribution is in the convex hull of the source label distributions, $\mathrm{Conv}(S)$. Then, we can find a convex combination $\omega$ of source distributions $S_i(Y)$ that recreate $T(Y)$, that is, $T(Y) = \sum_{i=1}^{M} \omega_i S_i(Y)$. However, when there are more clients than labels, $M > K$, such a *satisfying combination* $\omega$ need not be unique and different combinations may have different effective sample size. Let $\Omega = \{\omega \in \Delta^{M-1} : T(Y) = \omega^\top S(Y)\}$ denote all satisfying combinations where $S(Y) \in \mathbb{R}^{M \times K}$ is the matrix of all client label marginals. For a sufficiently small regularization penalty $\lambda$, the solution to equation 8 will be the satisfying combination with largest effective sample size.

$$\lim_{\lambda \to 0} \alpha^\lambda = \arg\min_{\alpha \in \Omega} \sum_{i=1}^{M} \frac{\alpha_i^2}{n_i} \ .$$

If there are fewer clients than labels, $M < K$, the set of target distributions for which a satisfying combination exists has measure zero, see Figure 1 (left). Nevertheless, the two cases above allow us to interpolate between being as faithful as possible to the target label distribution ($\lambda \to 0$) and retaining the largest effective sample size ($\lambda \to \infty$), the latter coinciding with FedAvg. Finally, when $T \in \text{Conv}(S)$ and $\lambda \to 0$, Proposition 1 applies also to FedPALS; the aggregation strategy results in an unbiased estimate of the target risk gradient in the SGD setting. However, like the unregularized weights, Proposition 1 does not apply for multiple local client updates.

**Case 3:** $T \notin \text{Conv}(S)$  If the target distribution does not lie in $\text{Conv}(S)$, see Figure 1 (left), FedPALS projects the target to the "closest point" in $\text{Conv}(S)$ if $\lambda = 0$, and to a tradeoff between this projection and the FedAvg aggregation if $\lambda > 0$. We have a discussion on how to choose $\lambda$ in the second and third cases in Appendix B.3.

### B.3   Choice of hyperparameter $\lambda$

A salient question in Cases 2 & 3, as discussed in Section 3.2, is how to choose the strength of the regularization, $\lambda$. A larger value will generally favor influence from more clients, provided that they have sufficiently many samples. When $T \notin \text{Conv}(S)$, the convex combination closest to $T$ could have weight on a single vertex. This will likely hurt the generalizability of the resulting classifier. In experiments, we compare values of $\lambda$ that yield different effective sample sizes, such as 10%, 25%, 50% or 75% of the original sample size, $N$. We can find these using binary search by solving equation 8 and calculate the ESS. One could select $\lambda$ heuristically based on the the ESS, or treat $\lambda$ as a hyperparameter and select it using a validation set. Although this would entail training and evaluating several models which can be seen as a limitation. We elect to choose a small set of $\lambda$ values based on the ESS heuristic and train models for these. Then we use a validation set to select the best performing model. This highlights the usefulness of the ESS as a heuristic. If it is unclear which values to pick, one could elect for a simple strategy of taking the ESS of $\lambda = 0$ and 100% and taking $l$ equidistributed values in between the two extremes, for some small integer $l$.

### B.4   Sampling strategies

Sparsity sampling entails randomly removing a subset of labels from clients and the target domain following the data set partitioning technique introduced in McMahan et al. (2017). Each client is assigned $C$ random labels, with an equal number of samples for each label and no overlap among clients. Sparsity sampling has been extensively used in subsequent studies (Geyer et al., 2017; Li et al., 2020a; 2022).

Dirichlet sampling simulates non-i.i.d. label distributions by, for each client $i$, drawing a sample $p_i \sim \text{Dirichlet}_K(\beta)$, where $p_i(k)$ represents the proportion of samples in client $i$ that have label $k \in [K]$. The concentration parameter $\beta > 0$ controls the sparsity of the client distributions. In dirichlet sampling, using a smaller $\beta$ results in more heterogeneous client data sets, while a larger value approximates an i.i.d. setting. This widely-used method for sampling clients was first introduced by Yurochkin et al. (2019).

### B.5   Oracle construction

The *Oracle* method serves as a benchmark to illustrate the performance upper bound when there is no distribution shift between the clients and the target. To construct this *Oracle*, we assume that the client label distributions are identical to the target label distribution, effectively eliminating the label shift that exists in real-world scenarios.

In practice, this means that for each dataset, the client data is drawn directly from the same distribution as the target. The aggregation process in the *Oracle* method uses FedAvg, as no adjustments for label shift are needed. Since the client and target distributions are aligned, FedPALS would behave equivalently to FedAvg under this setting, as there is no need for reweighting the client updates.

This method allows us to assess the maximum possible performance that could be achieved if the distributional differences between clients and the target did not exist. By comparing the *Oracle* results to those of

| $\delta$ | Accuracy |
|---|---|
| $10^{-3}$ | 88.8 |
| $10^{-2}$ | 85.2 |
| $5 \times 10^{-1}$ | 81.4 |

Table 2: Results of perturbing $\mathcal{T}$ with varying noise levels $\delta$.

our proposed method and other baselines, we can highlight the impact of label shift on model performance and validate the improvements brought by FedPALS.

### B.6 Perturbation of target marginal $\mathcal{T}$

In an experiment we perturb the given target label marginals, $\mathcal{T}$, to evaluate the performance impact of noise in the estimate. We do this by generating gaussian noise, $\epsilon$, and then we add the noise to the label marginal to create a new target $\mathcal{T}_p$. We modulate the size of the noise with a parameter $\delta$ and only add the positive noise values.

$$\mathcal{T}_p = \mathcal{T} + \delta \max(\epsilon, 0)$$

This is then normalised and used as the new target label marginal. This perturbation was done on the PACS experiment with $\delta \in [10^{-3}, 10^{-2}, 5 \times 10^{-1}]$ and repeated for three seeds. The results are given in Table 2 where we see that the performance decreases with increasing noise.

### B.7 Synthetic task

We randomly sampled three means $\mu_1 = [6, 4.6], \mu_2 = [1.2, -1.6]$, and $\mu_3 = [4.6, -5.4]$ for each label cluster, respectively.

### B.8 PACS

In this task we use the official source and target splits which are given in the work by Bai et al. (2024). We construct the task such that the training data is randomly assigned among three clients, then we remove the samples of one label from each of the clients. This is chosen to be labels '0', '1' and '2'. Then the client that is missing the label '2' is reduced so that it is 10% the amount of the original size. For the target we modify the given one by removing the samples with label '2', thereby making it more similar to the smaller client. To more accurately reflect the target distribution we modify the source domain validation set to also lack the samples with label '2'. This is reasonable since we assume that we have access to the target label distribution.

We pick four values of $\lambda$, [0,12,42,93], which approximately correspond to an ESS of $15\%, 25\%, 50\%$ and $75\%$ respectively. We use the same hyperparameters during training as Bai et al. (2024) report using in their paper. Furthermore, we use the cross entropy loss in this task.

### B.9 iWildCam

We perform this experiment using the methodology described in Bai et al. (2024) with the heterogeneity set to the maximum setting, i.e., $\lambda = 0$ in their construction.[2] We use the same hyperparameters which is used for FedAvg in the same work to train FedPALS. We perform 80 rounds of training and, we then select the best performing model based on held out validation performance and report the mean and standard deviation over three random seeds. This can be seen in Table 3. We pick four values of $\lambda$, [0,600,2500,5800], which approximately correspond to an ESS of $8\%, 25\%, 50\%$ and $75\%$ respectively. We use the cross entropy loss in this task.

Due to FedProx performing comparably to FedPALS on this task, in contrast with other experiments, we also perform an experiment where we do both FedProx and FedPALS. This is easily done as FedProx is a

---

[2]Note that this is not the same $\lambda$ used in the trade-off in FedPALS.

client side method while FedPALS is a weighting method applied at the server. This results in the best performing model.

We use the same hyperparameters during training as Bai et al. (2024) report using in their paper. However, we set the amount of communication rounds to 80.

## C  Additional empirical results

Figure 4 illustrates the aggregation weights of clients in the iWildCam experiment for $\lambda$ corresponding to different effective sample sizes.

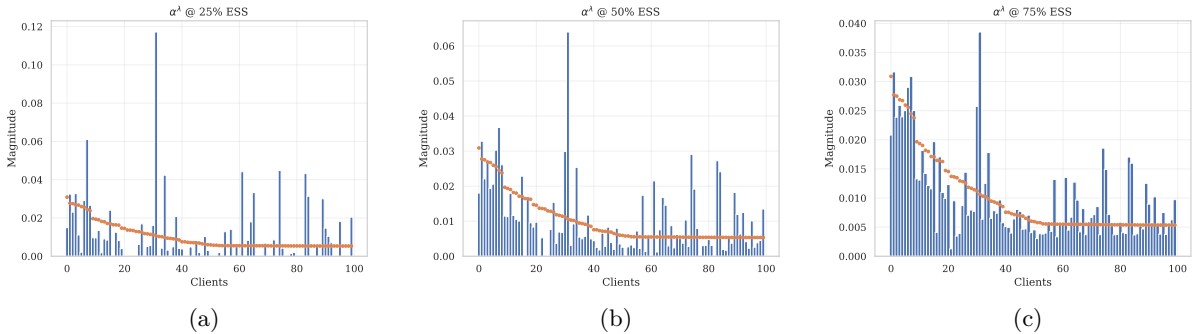

Figure 4: An illustration of the aggregation weights of clients in the iWildCam experiment using FedPALS for different ESS. The clients are sorted by amount of samples in descending order. The magnitude of the weights produced by federated averaging is shown as dots. Note that with increasing the ESS, the magnitudes more closely resemble that of federated averaging.

We report the performance of the models selected using the held out validation set in Table 3 and Table 4 for the iWildCam and PACS experiments respectively.

Table 3: Results on iWildCam with 100 clients, standard deviation reported over 3 random seeds.

| Algorithm | F1 (macro) |
|---|---|
| **FedPALS,** $\lambda = 0$ | $0.13 \pm 0.00$ |
| **FedPALS,** $\lambda = 600$ | $0.18 \pm 0.00$ |
| **FedPALS,** $\lambda = 2500$ | $0.19 \pm 0.00$ |
| **FedPALS,** $\lambda = 5800$ | $0.21 \pm 0.00$ |
| **FedProx+FedPALS,** $\lambda = 5800$ | $0.23 \pm 0.00$ |
| **FedAvg** | $0.20 \pm 0.01$ |
| **FedProx** | $0.21 \pm 0.00$ |
| **SCAFFOLD** | $0.15 \pm 0.01$ |
| **AFL** | $0.005 \pm 0.0$ |

### C.1  Results on CIFAR-10 with Dirichlet sampling

Figure 5 shows the results for the CIFAR-10 experiment with Dirichlet sampling of client and target label distributions.

### C.2  TRAINING DYNAMICS FOR FASHION-MNIST

Figure 7 shows the training dynamics for Fashion-MNIST and CIFAR-10 with different label marginal mechanisms.

Table 4: Results on PACS with 3 clients with mean and standard deviation reported over 3 random seeds.

| Algorithm | Accuracy |
|---|---|
| **FedPALS,** $\lambda = 0$ | $86.0 \pm 2.9$ |
| **FedPALS,** $\lambda = 12$ | $84.3 \pm 2.5$ |
| **FedPALS,** $\lambda = 42$ | $81.7 \pm 1.2$ |
| **FedPALS,** $\lambda = 93$ | $77.3 \pm 1.6$ |
| **FedProx+FedPALS,** $\lambda = 0$ | $87.2 \pm 4.1$ |
| **FedAvg** | $73.4 \pm 1.6$ |
| **FedProx** | $75.3 \pm 1.3$ |
| **SCAFFOLD** | $73.9 \pm 0.3$ |
| **AFL** | $74.5 \pm 0.9$ |

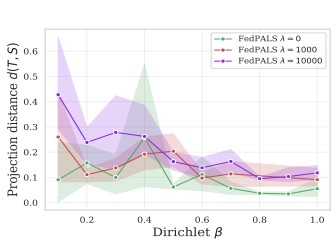

(a) Projection distance between target label marginal and client convex hull

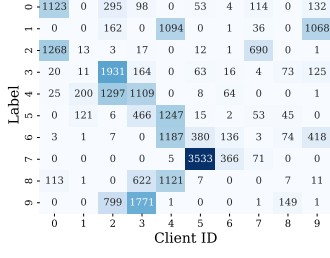

(b) Marginal label distributions of clients and target with $\beta = 0.1$.

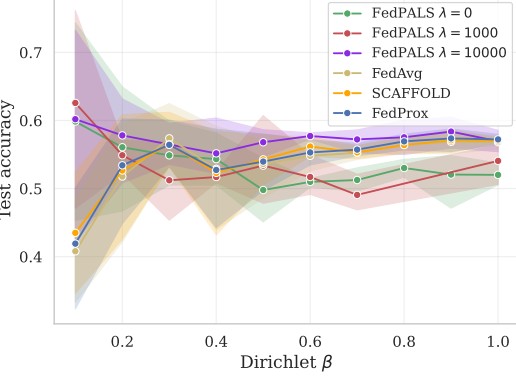

(c) Accuracy under Dirichlet sampling.

Figure 5: Results on CIFAR-10 with Dirichlet sampling across 10 clients, varying concentration parameter $\beta$. Clients with IDs 0–8 are clients present during training, and client with ID 9 is the target client.

## C.3 Local epochs and number of clients

In Figure 8c we show results for varying number of clients for each method. For the cases with number of clients 50 and 100, we use the standard sampling method of federated learning where a fraction of 0.1 clients are sampled in each communication round. In this case, we optimize $\alpha^\lambda$ for the participating clients in each communication round. Interestingly, we observe that while FedAvg performs significantly worse than FedPALS on a target client under label shift, it outperforms both FedProx and SCAFFOLD when the number of local epochs is high ($E = 40$), as shown in Figure 8b.

## C.4 Synthetic experiment: effect of projection distance on test error

When the target distribution $T(Y)$ is not covered by the clients, FedPALS finds aggregation weights corresponding to a regularized projection of $T$ onto $\mathrm{Conv}(S)$. To study the impact of this, we designed a controlled experiment where the distance of the projection is varied. We create a classification task with three classes, $\mathcal{Y} = \{0, 1, 2\}$, and define $p(X \mid Y = y)$ for each label $y \in \mathcal{Y}$ by a unit-variance Gaussian distribution $\mathcal{N}(\mu_y, I)$, with randomly sampled means $\mu_y \in \mathbb{R}^2$. We simulate two clients with label distributions $S_1(Y) = [0.5, 0.5, 0.0]^\top$ and $S_2(Y) = [0.5, 0.0, 0.5]^\top$, and $n_1 = 40$, $n_2 = 18$ samples, respectively. Thus, FedAvg gives larger weight to Client 1. We define a target label distribution $T(Y)$ parameterized by $\delta \in [0, 1]$ which controls the projection distance $d(T, S)$ between $T(Y)$ and $\mathrm{Conv}(S)$,

$$T_\delta(Y) \coloneqq (1 - \delta)T_{\mathrm{proj}}(Y) + \delta T_{\mathrm{ext}}(Y) \,,$$

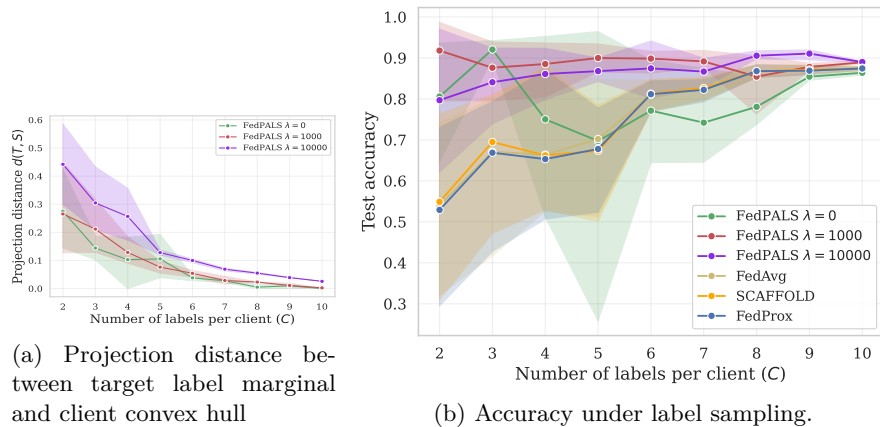

(a) Projection distance between target label marginal and client convex hull

(b) Accuracy under label sampling.

Figure 6: Results on Fashion-MNIST with label sampling across 10 clients, varying parameter $C$. Clients with IDs 0–8 are clients present during training, and client with ID 9 is the target client.

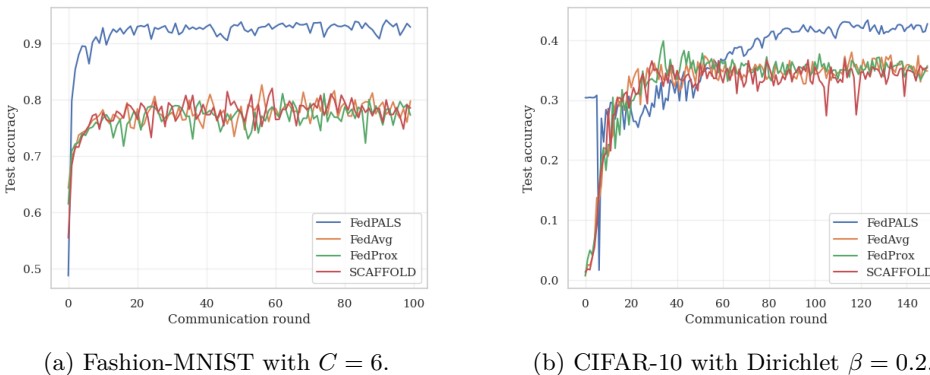

(a) Fashion-MNIST with $C = 6$.

(b) CIFAR-10 with Dirichlet $\beta = 0.2$.

Figure 7: Test accuracy during training rounds.

with $T_{\text{ext}}(Y) = [0, 0.5, 0.5]^\top \notin \text{Conv}(S(Y))$ and $T_{\text{proj}}(Y) = [0.5, 0.25, 0.25]^\top \in \text{Conv}(S(Y))$. By varying $\delta$, we control the projection distance $d(T, S)$ between each $T_\delta$ and $\text{Conv}(S)$ from solving equation 7, allowing us to study its effect on model performance.

We evaluate the global model on a test set with $n_{\text{test}} = 2000$ samples drawn from the target distribution $T(Y)$ for each value of $\delta$ and record the target accuracy for FedPALS and FedAvg. Figure 9 illustrates the relationship between the target accuracy and the projection distance $d(T, S)$ due to varying $\delta$. When $d(S, T) = 0$ (i.e., $T(Y) \in \text{Conv}(S)$), the target accuracy is highest, indicating that our method successfully matches the target distribution. As $d(S, T)$ increases (i.e., $T$ moves further away from $\text{Conv}(S)$), the task becomes harder and accuracy declines. For all values, FedPALS performs better than FedAvg. For more details on the synthetic experiment, see Appendix B.

# D   Proofs

## D.1   FedPALS updates

**Proposition 1 (Repeated)** (Unbiased SGD update)**.** Consider a single round $t$ of federated learning in the batch stochastic gradient setting with learning rate $\eta$. Each client $i \in [M]$ is given parameters $\theta_t$ by the server, computes their local gradient, and returns the update $\theta_{i,t} = \theta_t - \eta \nabla_\theta \hat{R}_i(h_{\theta_t})$. Let weights $\omega$ satisfy $T(X, Y) = \sum_{i=1}^M \omega_i S_i(X, Y)$. Then, the aggregate update $\theta_{t+1} = \sum_{i=1}^M \omega_i \theta_{i,t}$ satisfies

$$\mathbb{E}[\theta_{t+1} \mid \theta_t] = \mathbb{E}[\theta_{t+1}^T \mid \theta_t] \ ,$$

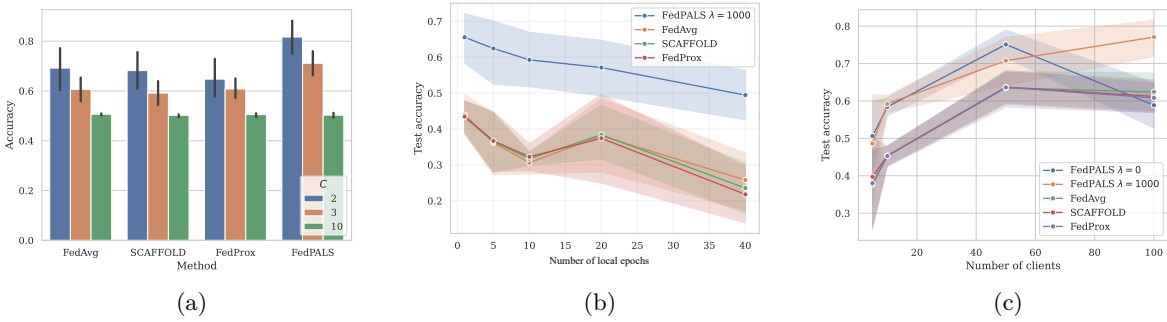

(a)  (b)  (c)

Figure 8: Comparison of CIFAR-10 results with different clients and settings. (a) 100 clients for $C = 2, 3, 10$, $\lambda = 1000$. (b) 10 clients and number of labels $C = 3$. We plot test accuracy as a function of number of local epochs $E$. The total number of communication rounds $T$ are set such that $T = E/150$, where 150 is the number of rounds used for $E = 1$. (c) Test accuracy as a function of number of clients, with $C = 3$.

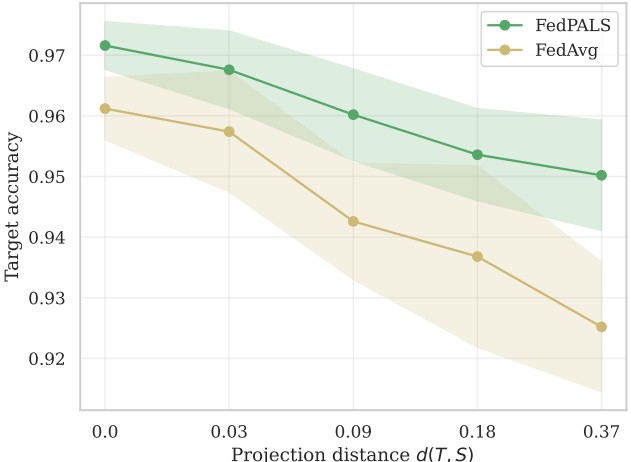

Figure 9: Synthetic experiment. Accuracy of the global model as a function of the projection distance $d(T, S)$ between the target distribution $T(Y)$ and client label distributions $\text{Conv}(S(Y))$. Means and standard deviations reported over 5 independent runs.

where $\theta_{t+1}^T$ is the batch stochastic gradient update for $\hat{R}_T$ that would be obtained with a sample from the target domain.

*Proof.*

$$\theta_{t+1} = \sum_{i=1}^{M} \omega_i \theta_{i,t} = \sum_{i=1}^{M} \omega_i (\theta_t - \eta \nabla \hat{R}_i (h_{\theta_t})) = \theta_t - \eta \sum_{i=1}^{M} \omega_i \nabla \hat{R}_i (h_{\theta_t}) \tag{11}$$

$$\mathbb{E}[\theta_{t+1} \mid \theta_t] = \theta_t - \eta \cdot \mathbb{E}\left[\sum_{i=1}^{M} \omega_i \nabla \hat{R}_i(h_{\theta_t}) \mid \theta_t\right] \tag{12}$$

$$= \theta_t - \eta \cdot \sum_{x,y} \mathbb{E}\left[\sum_{i=1}^{M} \hat{S}_i(x,y)\omega_i\right] \nabla L(y, h_{\theta_t}(x)) \tag{13}$$

$$= \theta_t - \eta \cdot \sum_{x,y} T(x,y) \nabla L(y, h_{\theta_t}(x)) \tag{14}$$

$$= \theta_t - \eta \cdot \mathbb{E}\left[\sum_{x,y} \hat{T}(x,y)\right] \nabla L(y, h_{\theta_t}(x)) = \mathbb{E}[\theta_{t+1}^T \mid \theta_t] . \tag{15}$$

$\square$

### D.2  FedPALS in the limits

As $\lambda \to \infty$, because the first term in equation 8 is bounded, the problem shares solution with

$$\min_{\alpha_1,\dots,\alpha_M} \sum_i \frac{\alpha_i^2}{n_i} \quad \text{s.t.} \quad \sum_i \alpha_i = 1, \quad \forall i : \alpha_i \geq 0 . \tag{16}$$

Moreover, we have the following result.

**Proposition 4.** *The optimization problem*

$$\min_{\alpha} \sum_i \frac{\alpha_i^2}{n_i} \quad s.t \quad \sum_i \alpha_i = 1 \quad \alpha_i \geq 0 \; \forall \; i \; ,$$

*has the optimal solution* $\alpha_i^* = \frac{n_i}{\sum_i n_i}$ *where* $i \in [1, m]$

*Proof.* From the constrained optimization problem we form a Lagrangian formulation

$$\mathcal{L}(\alpha, \mu, \tau) = \sum_i \frac{\alpha_i^2}{n_i} + \underbrace{\mu\left(1 - \sum_i \alpha_i\right)}_{h(\alpha)} + \tau \underbrace{-\alpha}_{g(\alpha)}$$

We then use the KKT-theorem to find the optimal solution to the problem.

$$\nabla_\alpha \mathcal{L}(\alpha^*) = 0 \implies \forall i : \quad 2\frac{\alpha_i^*}{n_i} - \mu - \tau = 0 . \tag{17}$$

In other words, the following ratio is a constant,

$$\forall i \quad \frac{\alpha_i^*}{n_i} = c$$

for some constant $c$. We have the additional conditions of primal feasibility, i.e.

$$h(\alpha^*) = 0$$
$$g(\alpha^*) \leq 0$$

From the first constraint, we have $\sum_{i=1}^{M} \alpha_i^* = 1$, and thus,

$$\sum_{i=1}^{M} \alpha_i^* = c \sum_{i=1}^{M} n_i = 1$$

which implies that $c = 1/\sum_{i=1}^{M} n_i$ and thus

$$\forall i : \alpha_i^* = \frac{n_i}{\sum_{i=1}^{M} n_i} .$$

$\square$

