# OpenReview forum: "Overcoming label shift in target-aware federated learning"
_TMLR — Accepted by TMLR_

### Review · Reviewer_e8dL · 2026-04-05

**Summary Of Contributions:**

This paper studies the setting of federated learning under a shift in the label distribution. In particular, the paper points out that prior works on federated learning implicitly assumes that the target distribution is a simple aggregation of the client distributions, which may not be an always valid assumption. This paper relaxes the assumption and assumes that the target distribution lies in the convex hull of the client distributions, and that conditioned on the labels, the input distrubtions are the same. Under this assumption, the paper proposes the FedPALS algorithm that explicit matches the aggregated distribution to the target distribution. Theoretically, the paper shows that a properly chosen aggregation parameteter, the update of FedPALS matches the oracle update on the target distribution. Experimentally, the paper demonstrated better performance compared to the baseline on three federated learning datasets with label distribution shift.

**Audience:**

Yes

**Audience Explanation:**

The topic of federated learning is important in the field of ML. The incorporation of a label distribution shift seems an important consideration in order for algorithms to be applied to real-world scenarios.

**Broader Impact Concerns:**

None.

**Claims And Evidence:**

Yes

**Claims Explanation:**

The paper presents sufficient evidence for both the theoretical claim and empirical claims.

**Requested Changes:**

1. The paper assumes that the central server have access to the real label distribution. and that the target label distribution lies in a convex hull of the client label distributions. These two assumptions need to be better justified.
2. The papper need to discuss how to solve Eq. (8) in the implementation of the algorithm, and what is the complexity. Since the algorithm requires this inner optimization, reporting of wall-clock time in addition to communication rounds is also necessary.
3. I don't fully get why sparse label distirbution in the client can result in a larger projection distance in Figure 2(a). A clearer explanation might be needed.
4. It seems that the experiments are restricted to classification tasks. The paper should include results or at least discuss how to apply the algorithm to regression tasks.

---

> ### Author Response · Authors · 2026-04-08
> **Review response**
>
> Dear reviewer e8dL,
> Thank you for your comments and requested changes to the paper.
> We have uploaded a revised version of the paper, and provided answers below.
>
> ### 1. Assumptions
> We do *not* assume that the target label distribution lies in the convex hull in general. Equations (7) and (8) are aimed at precisely the case where it does not hold. Look for the paragraph before equation (7) which starts with ``In applications, the target may not be covered by clients...''. We have clarified this in our revision by adding a new section heading dedicated to the non-covered case.
>
> We discuss the availability of the target distribution to the server extensively, with examples, on page 3. Our paper sets out to answer the question: If the server knows the label distribution where the federated model will be used, how can this be exploited to yield better performance? We believe that our results demonstrate the value of this question.
>
> ### 2. Complexity
> The problem in (8) is an entirely standard quadratic program (QP) with $M$ variables, one per client, and a simplex constraint. There are many off-the-shelf solvers that can solve it, such as SciPy, Gurobi, CVX, etc. The time to solve the problem is entirely negligible in the context of federated learning as (i) it typically takes less than a second, (ii) it is done *once*, before federated learning starts. With general-purpose methods, solving a simplex-constrained QP is $O(M^3)$ in the worst case, but we stress that this will never dominate the time for training the model.
>
> ### 3. Effect of sparse label distributioin
> This is because the target distribution is also more sparse when $C$ is lower. We have clarified this in the figure caption. When this happens, other clients will have support on labels that are unsupported in the target, and no matter how you combine them, there the resulting combination will have support on a label where the target has probability 0. This phenomenon is explained in the paragraphs leading up to equation (7). We have expanded this as well.
>
> ### 4. Generalization to regression problems
> Our focus in this work is specifically on label shift in multi-class classification, as we specify in Section 2. While extending our method to regression is an interesting direction, it requires fundamentally different mathematical machinery, placing it outside the scope of this paper.
>
> In our classification setting, the label space is discrete ($Y \in \{1, \dots, K\}$). This allows the target and client label distributions to be exactly represented on a discrete probability simplex, which is why we can successfully formulate the target-aware aggregation as the discrete Quadratic Program seen in Equation (8).
>
> In contrast, addressing label (target) shift in regression involves a continuous target space ($Y \in \mathbb{R}$). Aligning distributions in a continuous setting requires estimating continuous density ratios ($T(Y)/S(Y)$) using techniques such as Kernel Density Estimation (KDE) or specialized density ratio estimators, rather than computing Euclidean projections onto a discrete simplex. Because this fundamentally shifts the mathematical approach from discrete weight optimization to continuous density estimation, adapting FedPALS for regression would constitute an entirely separate study.

---

> ### Comment · Action_Editor_sxDJ · 2026-05-22
>
> Dear Reviewer e8dL,
>
> When you have some time, can you take a look at the rebuttal and submit your official recommendation? Thank you!
>
> AE

---

### Review · Reviewer_Xxz1 · 2026-04-20

**Summary Of Contributions:**

The paper explores and proposes a solution for a federated learning setup, where the final model has to generalize to a target domain whose label distribution is different from the clients and the aggregate model (FedAvg). The authors propose a server aggregation method where a quadratic program is used to finds weights to aggregate the model, such that the learnt weights best approximate the target label marginal. The clients share the label marginal before the training process and the server learns the weights.

Strengths:
The paper is well written and proposes a simple solution to an underexplored problem. Although the paper toggles between empirical and theoretical analysis, the authors sufficiently support the claims made.

Weakness:
1. The motivation of the federated setting is a bit unclear. Is this a realistic federated learning setup, where clients participate in a federated setup, where the server is optimizing for a target that is different from most clients, while those clients are donating their data to train a model, risking privacy as well?
2. Experimentation is limited to 3-10 clients which is very few and raises doubts on the efficacy of the method when a large set of clients participate.
3. The cost of sharing client marginals is underexplored in this paper. There are no privacy analysis or experiments done.
4. Selection of lambda values require training 4 separate models and selecting via a validation set from the target distribution. Training 4 models could be feasibile when you have 2-3 clients, but not at scale. Does this also not violate the assumption, that no target distribution samples are available?

**Audience:**

Yes

**Audience Explanation:**

Overall, the paper proposes a simple mechanism in an interesting federated setting.

**Broader Impact Concerns:**

No ethical concerns.

**Claims And Evidence:**

No

**Claims Explanation:**

Overall the claims made by the paper are well supported with theoritical or empirical evidence. I would like to get clarification on the following:
1. Low clients to deem good empirical performance: The paper uses 10 clients for MNIST and 3 clients for PACS, which is quite low for federated learning experiments. It is hard to judge the amount of improvement the proposed method brings in with this few clients and seeds (3 for PACS), especially when there are no statistical significance tests provided and given the high standard deviations. It is also not clear, as with more clients, if the improvements still stand.

2. Selection of lambda values require training 4 separate models and selecting via a validation set from the target distribution. Training 4 models could be feasibile when you have 2-3 clients, but not at scale. Does this also not violate the assumption, that no target distribution samples are available?

**Requested Changes:**

The paper would be strengthened and ready for acceptance, if the authors could provide large scale experiments across 100s of clients and demonstrate the efficacy of the proposed method, along with statistical significance.

---

> ### Author Response · Authors · 2026-05-04
> **Late rebuttal**
>
> Dear Reviewer Xxz1,
>
> Thank you for your questions and constructive feedback.
> We apologize for responding to your review late. Hopefully, you will still take this response into consideration in your recommendation.
>
> **With regards to what the points you would like clarifications on, and suggested changes:**
> 1. We do perform experiments on iWildCam with 100 clients, as is highlighted in Figure 3(b). We did not include the results for this in Table 1 since several of the baseline methods did not run or produced meaningless results for this data set. But the results are available in Appendix Table 3.
> 2. Yes, selecting $\lambda$ through performance on a held-out validation set can be challenging, especially if no such validation set is available. This is precisely why we suggest a rule of thumb for getting started based on the effective sample size in Appendix B.3. We will extend our discussion to cover this issue.
>
> Sincerely,
>
> The authors

---

> ### Comment · Action_Editor_sxDJ · 2026-05-22
>
> Dear Reviewer Xxz1,
>
> When you have some time, can you take a look at the rebuttal and submit your official recommendation? Thank you!
>
> AE

---

### Review · Reviewer_jXGc · 2026-04-21

**Summary Of Contributions:**

This work proposes a novel framework to train federated models under the label shift. Specifically, authors propose a novel aggregation schema which uses the known label distributions to permit the performance on the target domain (known only to the central server). Authors empirically demonstrate the effectiveness of the method in comparison to prior baselines in the field.

**Audience:**

Yes

**Audience Explanation:**

The work itself would be of interest to the community, while the whole field of non-IID FL, label shift etc. has been previously been well-explored, the specific problem this work solves is an interesting one. How do the existing frameworks compete when the target domain (of the central server) is not covered by the client label distributions. The idea is quite simple, and while I have some concerns and comments, it makes sense to me and the work showcases its effectiveness through experimental results.

My first small concern is the positioning wrt the rest of FL literature. Here it is clear to me that the focus of the work is much more about utility-preserving aspects of cross-silo FL, which makes sense wrt experimental design. I do think that some other aspects of FL relevant to this setup have not been covered in enough details (in text, I do not expect any additional experimental results): how is this problem different to an incomplete participation round in cross-device FL, for instance? Do any additional hyperparameters affect the effectiveness or is it just the label/data split that was experimentally evaluated?

My second small concern is that while it is a utility-oriented work, there was nothing discussed wrt the privacy implications of the method. Authors added a few lines in the introduction as part of their response to another reviewer that describe 'privacy' as one of the potential problems FL participants are facing. I was, therefore, surprised there were no discussion points/related works/limitations related to this and compatibility with any privacy-enhancing technologies frequently used in FL like MPC or differential privacy. On the latter: I would actually encourage the authors to discuss the implications of DP in this setting. To me it seems a relatively easy future extension in the context of sharing privatised versions of label distributions with the central server instead of the original ones (based on what is implied by the 'label distribution' that is being sent to the central server, I was not 100% certain I understood the format of data being shared). Either way, I do not expect this as 'experimental evidence', but I would like the authors to add some thoughts on the privacy implications of their method (as well as mitigations) as part of limitations/discussion/future work.

**Broader Impact Concerns:**

No ethical implications to discuss.

**Claims And Evidence:**

Yes

**Claims Explanation:**

There is not much for me to add here, the experiments are sound, the results are promising, the algorithm is clear to me. Nothing to add to this part except for a minor note on what the shared label distribution actually looks like (maybe a small example similar to the one on the different disjoint labels authors already had in the text). The reason is discussed in the section below.

**Requested Changes:**

As mentioned above:
- Short example of what the shared label distribution data looks like.
- Discussion on the similarities/differences with previously studied FL problems related to label shift due to partial participation etc.
- Discussion on the privacy implications and potential mitigations against honest-but-curious or malicious central servers (and how such mitigations can limit, if applicable, the effectiveness of the overall training algorithm).

---

> ### Author Response · Authors · 2026-05-04
> **Late rebuttal**
>
> Dear Reviewer jXGc,
>
> Thank you for your thoughtful and constructive review!
>
> We apologize for responding to your review late. Hopefully, you will still take this response into consideration in your recommendation.
>
> **With regards to your suggested changes:**
> 1. Figures 2b and 5b show examples of the label distribution of the target (Client ID 9 is the target—we can make this more clear!). Or did you mean the average distributions over clients?
> 2. We would be happy to include a discussion about previous FL studies related to label shift in the camera-ready version.
> 3. Similarly, we will discuss, in the camera-ready version, honest-but-curious and malicious servers, while being clear that we are assuming a neutral or trustworthy server in our study.
>
> Thank you for these suggestions for how to improve our submission!
>
> Sincerely,
>
> The authors

---

> > ### Comment · Reviewer_jXGc · 2026-05-05
> > **Response**
> >
> > Dear authors,
> >
> > Thank you for your response.
> >
> > Re: points 2 and 3 - ok.
> >
> > Re: point 1 - I meant what does the server actually see about the 'label distribution'? Does this mean each client sends to the central server their respective class label histogram (or a map of label-count) etc.? It was no clear to me which specific information regarding the labels does the central server gets from each client (and also client 9 being the target was not obvious to me from the figure, so, please, do highlight that too).

---

> > > ### Author Response · Authors · 2026-05-05
> > > **Re:**
> > >
> > > Yes, that is correct! Each client sends the proportion of instances of each class to the server, the empirical marginal label distribution, equivalent to the class label histogram.
> > >
> > > We will relabel the Figure, so that it is clear what is the target!

---

### Decision · Action_Editor_sxDJ · 2026-06-02

**Recommendation:** Accept with minor revision

**Audience:**

Yes

**Audience Explanation:**

Federated learning is clearly relevant to TMLR.

**Claims And Evidence:**

Yes

**Claims Explanation:**

In general, the answer is of course yes. However, Reviewer Xxz1 commented that
> While the paper solves an important problem, **there are certain assumptions that are not clearly laid out:** 1) Practicality of training multiple models for hyperparameter tuning and 2) Experimentation on large number of clients is not clearly presented.

and Reviewer jXGc commented that
> Authors have addressed (**or promised to address**) all of my comments;

Therefore, I think "accept with minor revision" should be better, and please address all remaining issues in the final version.

---

> ### Author Response · Authors · 2026-06-29
> **Re:**
>
> Dear AE,
>
> We have now uploaded a camera-ready revision including names and affiliations and the fixes asked for above.
> We uploaded a normal revision first highllighting the relevant paragraphs in blue, for your convenience.
>
> Best,
> Authors